

# Mapping theme trends and recognizing hot spots in postmenopausal osteoporosis research: a bibliometric analysis

Siming Zhou*, Zhengbo Tao*, Yue Zhu and Lin Tao

Department of Orthopaedics, First Hospital of China Medical University, Shenyang, Liaoning, China
* These authors contributed equally to this work.

## ABSTRACT

**Background:** This study aimed to draw a series of scientific maps to quantitatively and qualitatively evaluate hot spots and trends in postmenopausal osteoporosis research using bibliometric analysis.

**Methods:** Scientific papers published on postmenopausal osteoporosis were extracted from the Web of Science Core Collection and PubMed database. Extracted information was analyzed quantitatively with bibliometric analysis by CiteSpace, the Online Analysis Platform of Literature Metrology and Bibliographic Item Co-Occurrence Matrix Builder (BICOMB). To explore the hot spots in this field, co-word biclustering analysis was conducted by gCLUTO based on the major MeSH terms/MeSH subheading terms-source literatures matrix.

**Results:** We identified that a total of 5,247 publications related to postmenopausal osteoporosis were published between 2013 and 2017. The overall trend decreased from 1,071 literatures in 2013 to 1,048 literatures in 2017. *Osteoporosis International* is the leading journal in the field of postmenopausal osteoporosis research, both in terms of impact factor score (3.819) and H-index value (157). The United States has retained a top position and has exerted a pivotal influence in this field. The University of California, San Francisco was identified as a leading institution for research collaboration, and Professors Reginster and Kanis have made great achievements in this area. Eight research hot spots were identified.

**Conclusions:** Our study found that in the past few years, the etiology and drug treatment of postmenopausal osteoporosis have been research hot spots. They provide a basis for the study of the pathogenesis of osteoporosis and guidelines for the drug treatment of osteoporosis.

Corresponding authors
Yue Zhu, zhuyuedr@163.com
Lin Tao, taolindr@163.com

## INTRODUCTION

Osteoporosis, described as the microstructural degeneration of bone tissue and low bone mass, is a systemic skeletal disease causing incremental bone fragility and sensitivity to fracture. There is an increasing incidence of osteoporotic fractures at all ages. Women have twice the risk of getting fractures as men, making postmenopausal osteoporosis, which

results from estrogen deficiency and leads to an increase in bone turnover, one of the most important types of primary osteoporosis. To repair micro-damage and adapt to mechanical and metabolic needs, bone is being continuously remodeled. The remodeling of bone is performed by two specialized cells: bone-forming osteoblasts and bone-resorbing osteoclasts (*Wu et al., 2015*). Additionally, the loss of connectivity in trabecular bone, and cortical bone thinning and loss of porosity are affected by an imbalance between bone formation and resorption. The existing treatment of osteoporosis is mainly drug-based. Diphosphonates are given as a firstline treatment, followed by denosumab (a RANKL inhibitor). Teriparatide (a fragment of parathyroid hormone) is the only approved anabolic agent. Estrogen replacement therapy or selective estrogen receptor modulators can be considered in specific conditions (*McClung et al., 2005*). The prevention of osteoporosis focuses on gaining maximum peak bone mass and minimizing postmenopausal and age-related bone loss through nutrition, maintenance of a normal body mass index, regular physical activity, and the absence of smoking (*Oncken et al., 2006*). By reducing falls in high-risk populations, fractures, the main complication of osteoporosis, may also be restrained (*Schwartz et al., 2005*).

In recent years, the bibliometric method used most often has been a quantitative analysis, which uses the statistical index to measure the contribution of a subject or scientific publications in an area of research, and shows widely-applied research trends and hot spots. While this method works to a certain extent, different scholars in this field have different results and views, and there is a lack of recent bibliometric research. French bibliometric scientists Callon et al. first presented the co-word analysis in 1986, which was utilized to find information and recognize hot spots in scholarly literature (*Hong et al., 2016*). To further summarize the focus of the research and structure of the subject by statistical analyses, such as factor analysis, cluster analysis, multivariate analysis or multidimensional scaling analysis, the significant keywords of a theme were categorized. Among these methods, cluster analysis has been widely used to extract a research theme area. Unlike conventional clustering, biclustering permits coinstantaneous cluster rows and columns of matrices, not just the global information, in order to efficiently detect local messages in high-dimensional data. The field of bibliometrics has recommended biclustering analysis in more recent years. *Fiannaca et al. (2015)* revealed miRNA expression profiles in breast cancer using biclustering (*Fiannaca et al., 2015*), and *Li et al. (2015*, *2016a*, *2016b)* applied biclustering to probe into subject areas and hot spots of research on Internet health information seeking behavior (*Zheng et al., 2015*). Their research findings suggested that the biclustering method can direct central research focus and the representative literature or research.

There have been few bibliometric studies on postmenopausal osteoporosis, and those few paid more attention to studying published information than future research trends (*Biglu, Ghavami & Biglu, 2014*; *Pluskiewicz et al., 2018*). In this study, an integrated analysis on the external features and content patterns of pertinent literature was performed to clarify the status and progress of postmenopausal osteoporosis research in the past five years. Particularly, co-word biclustering analysis was used to confirm the research hot

spots for postmenopausal osteoporosis. We hope that this research will provide some basis for future studies on postmenopausal osteoporosis.

## MATERIALS AND METHODS

### Data source and search strategy

Literatures were retrieved online through the Social Science Citation Index and the Science Citation Index-Expanded of the Web of Science Core Collection (WoSCC) on September 7, 2019. The search strategy was used for the following terms with a timeframe of the January 1, 2013 solstice to December 31, 2017: Osteoporosis, Postmenopausal AND Language = English, and only original articles and reviews were included. Related data were extracted and downloaded without the restriction of language from PubMed, developed by the National Center for Biotechnology Information (NCBI) of the National Library of Medicine (NLM), providing free access to MEDLINE, OLDMEDLINE, and other related databases. MeSH (Medical Subject Headings) terms are a series of standardized words that can map the content of literatures. According to the MeSH words used, co-word clustering analysis can be carried out continuously (*Li et al., 2015*). The search strategy applied was "Osteoporosis, Postmenopausal" (Mesh). Publication date was set from Jan 1st, 2013 to Dec 31st, 2017.

All of the literature retrieval and download recording were completed in the same day in order to reduce the quantity of citations resulting from frequent database updates.

### Data collection

Two investigators (Siming Zhou and Zhengbo Tao) independently conducted the primary search by screening the full text, titles and, in some cases, abstracts, of the literatures. The agreement rate between them was 0.90, showing a strong accordance (*Landis & Koch, 1977*). Before reaching an agreement, any differences were discussed. WoSCC data were converted to txt format and imported into CiteSpace V5.5.R1 SE, 64bit (Drexel University, Philadelphia, PA, USA) and the Online Analysis Platform of Literature Metrology (http://bibliometric.com/) for bibliometric analysis. Each downloaded literature was saved from PubMed as a file in XML format and imported into the Bibliographic Item Co-Occurrence Matrix Builder (BICOMB) (developed by Professor Cui from China Medical University and freely available online) (*Cui et al., 2008*) for hot spot analysis.

### Analysis methods

#### Bibliometric analysis

We tried to create "The WoSCC Literature Analysis Report" to summarize publication characteristics, such as journals, authors, countries, institution condition, number of annual publications, H index, and citation counts. To measure the scientific value of research, we enquired the Journal Citation Reports (JCR) 2018 to obtain the impact factor (IF) and the number of citations, which we regarded as important indicators (*Eyre-Walker & Stoletzki, 2013*). After evaluating these scientific metrics, it was easy to measure different aspects of the publications including their reputation, production and influence. In our study, we used the Literature Metrology online analysis platform to

analyze the annual number of publications and country/region growth tendencies. CiteSpace was used for collaboration network analysis to connect journals, authors, institutions and countries. CiteSpace can also use "time slicing", where you could set "years per slice" to 1 and set "top N per slice" to 50, and the top 50 papers in a 1-year slice would be extracted into a single network. According to the aim of our analysis, we selected different node types with the size representing citation counts or the number of publications. (*Chen, Ibekwe-SanJuan & Hou, 2010*; *Chen, 2006*).

### Co-word biclustering analysis of research hotspots

BICOMB and Microsoft Excel were utilized to identify the proportion of the frequency permutations of major MeSH terms/MeSH subheading terms in the concerned literature.

In this study, the tendencies of the extremely frequent major MeSH terms/MeSH subheading terms were visually stated. Meanwhile, in order to detect the hot spots of postmenopausal osteoporosis research, biclustering of the chosen publications and extremely frequent major MeSH terms/MeSH subheading terms was carried out. Biclustering was applied to show the relationship between source literatures and extremely frequent words, and the relationship among extremely frequent words. From BICOMB, a binary matrix with source literatures as the columns and extremely frequent major MeSH terms/MeSH subheading terms as the rows, was structured for further biclustering by means of the software "gCLUTO", version 1.0 (Graphical CLUstering TOolkit, a graphical front-end for the CLUTO data clustering library, developed by Rasmussen, Newman, and Karypis from the University of Minnesota) (*Karypis Lab, 2014*). Based on the literature, the parameters of biclustering in gCLUTO were set, and were suitable for biclustering analysis. I2 was then selected for criterion function, Cosine was chosen for similarity function, and repeated bisection for clustering method. The biclustering result of the matrix of source literatures showed extremely frequent major MeSH terms/MeSH subheading terms displayed by matrix visualization and mountain visualization. In order to identify the appropriate number of clusters, the biclustering with different numbers of clusters was redirected until the matrix visualization and mountain visualization reached the optimal result. With semantic relationships found between major MeSH terms/MeSH subheading terms and the typical source literatures in clusters, the fundamental structure of our research focus on postmenopausal osteoporosis was mapped and established.

## RESULTS

### Distribution characteristics of literature

#### Output of related literature

In total, 5,247 literatures comprising 4,466 articles and 781 reviews (Fig. 1), were involved in this study based on search strategy and inclusion criteria (Jan. 1st, 2013–Dec. 31st, 2017). The trend in the number of annual publications related to postmenopausal osteoporosis from 2013 to 2017 is shown in Fig. 2, where you can see the overall trend decreases from 1,071 literatures in 2013 to 1,048 literatures in 2017.

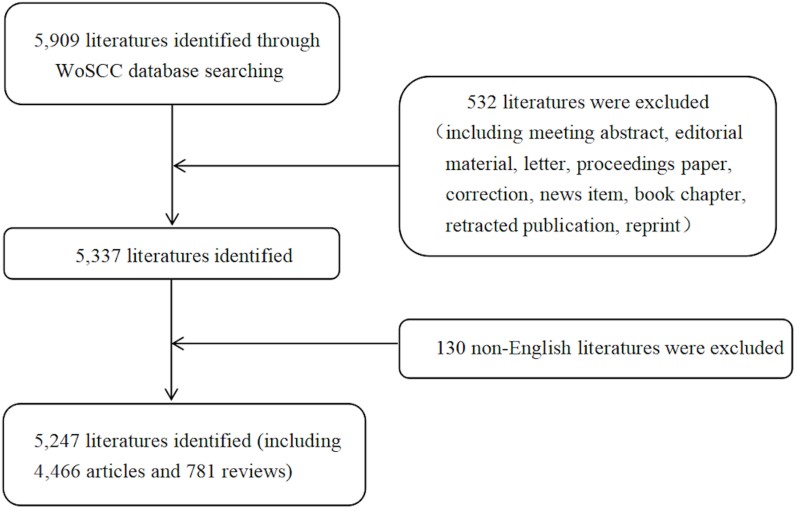

**Figure 1 Flow chart of literature filtering included in this study.**

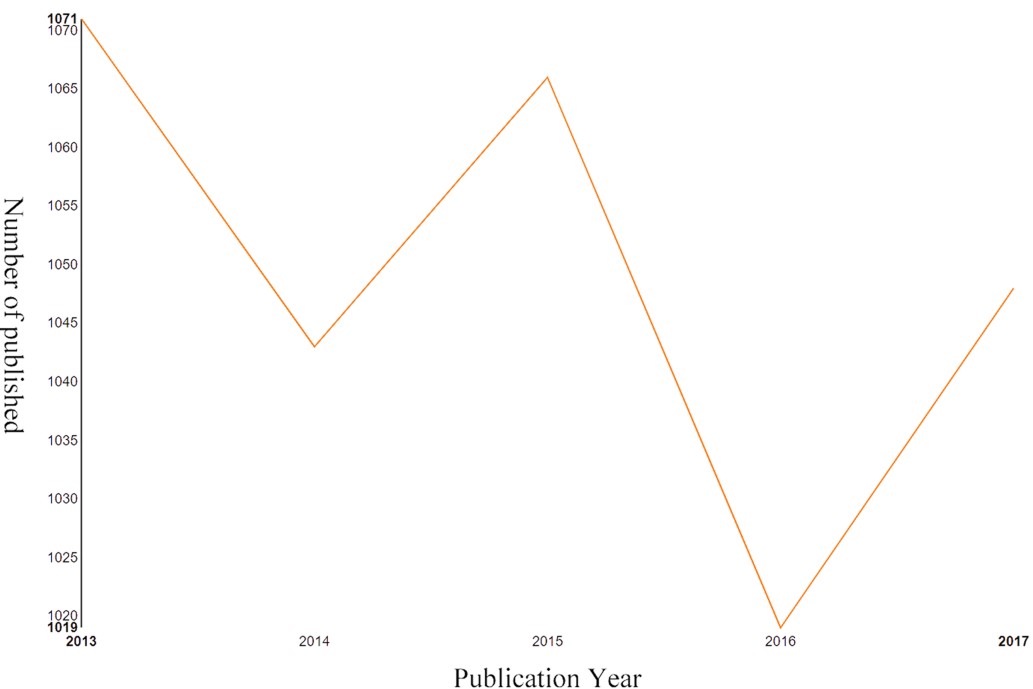

**Figure 2 Output of related literature.** The number of annual publications in postmenopausal osteoporosis from 2013 to 2017.

## Distribution characteristics of countries/regions and institutions

All of the literatures on postmenopausal osteoporosis contributed by active authors, based on rough statistics, stemmed from at least 81 different countries. The research findings on postmenopausal osteoporosis in different countries or regions are listed in Fig. 3. So far, the United States (1,378) has been the largest contributor to postmenopausal osteoporosis

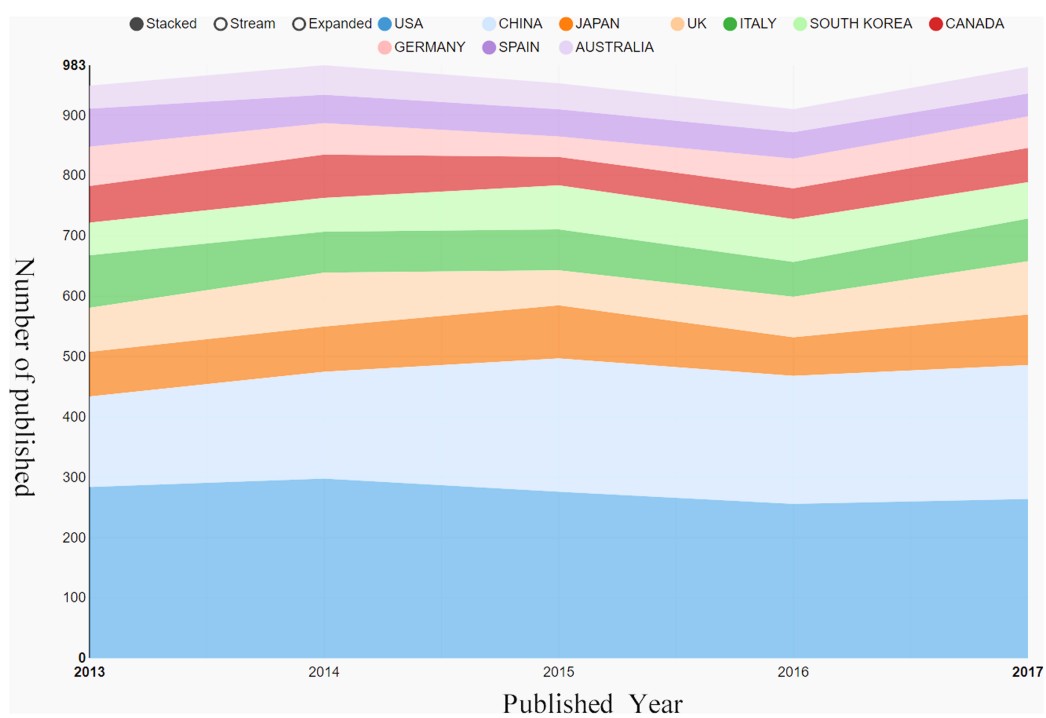

**Figure 3 Output of related literature.** The growth trends of the top 10 countries/regions in post-menopausal osteoporosis from 2013 to 2017.

research, followed by China (982), Japan (385), England (375), and Italy (352). In regards to the centrality index, although Spain's scientific research output was not very high, it had the largest influence on other countries (centrality = 0.14), followed by Australia (0.11) and the United States (0.10) (Table 1). The top 10 related research institutions ordered by the number of published papers included the University of California, San Francisco (131), Columbia University (129), Seoul National University (128), Amgen Inc (126), and Yonsei University (125) (Table 1). The postmenopausal osteoporosis research network map was a low-density map (density = 0.0843) (Fig. 4), implying that research groups were relatively dispersed across institutions, and that mutual cooperation still needs to be strengthened. Most centrality indexes were less than 0.15, demonstrating that the influence of most institutions is still at a low level and the amount of cooperation between institutions is inadequate. An analysis of international cooperation is shown in Fig. 5; the most frequent collaboration was between the United States and China, followed by the US and England.

## Most active journals

A total of 1,162 journals have recently emerged in this field. The 10 most active journals published 1,686 publications on postmenopausal osteoporosis, accounting for 32.13% of all 5,247 publications. The ranking of the top 10 active journals, which are recognized as the core journals in this field, is shown in Table 2. The top three journals are *Osteoporosis International*, *Bone*, and *Journal of Bone and Mineral Research*, and these three journals make up more than 18.27% of the entire indexed literatures in this area.

**Table 1 The top 10 countries/regions and institutions contributing to publications in postmenopausal osteoporosis research.**

| Rank | Country/ region | Article counts | Centrality | Institutions | Article counts | Centrality | Total number of citations | Average number of citations | Total number of first authors | Total number of first author citations | Average number of first author citations |
|---|---|---|---|---|---|---|---|---|---|---|---|
| 1 | US | 1378 | 0.10 | Univ Calif San Francisco | 131 | 0.08 | 1133 | 8.65 | 30 | 294 | 9.8 |
| 2 | People's Republic of China | 982 | 0.00 | Columbia Univ | 129 | 0.04 | 957 | 7.42 | 39 | 338 | 8.67 |
| 3 | Japan | 385 | 0.00 | Seoul Natl Univ | 128 | 0.03 | 203 | 1.59 | 35 | 72 | 2.06 |
| 4 | England | 375 | 0.02 | Amgen Inc | 126 | 0.05 | 1087 | 8.63 | 18 | 110 | 6.11 |
| 5 | Italy | 352 | 0.01 | Yonsei Univ | 125 | 0.15 | 172 | 1.38 | 30 | 33 | 1.1 |
| 6 | South Korea | 314 | 0.00 | Mayo Clin | 123 | 0.02 | 487 | 3.96 | 30 | 92 | 3.07 |
| 7 | Canada | 288 | 0.02 | Univ Sheffield | 115 | 0.16 | 1398 | 12.16 | 33 | 170 | 5.15 |
| 8 | Germany | 252 | 0.02 | Shanghai Jiao Tong Univ | 112 | 0.15 | 141 | 1.26 | 55 | 74 | 1.35 |
| 9 | Spain | 237 | 0.14 | Univ Toronto | 101 | 0.02 | 284 | 2.81 | 18 | 56 | 3.11 |
| 10 | Australia | 212 | 0.11 | Univ Liege | 93 | 0.02 | 1145 | 12.31 | 28 | 97 | 3.46 |

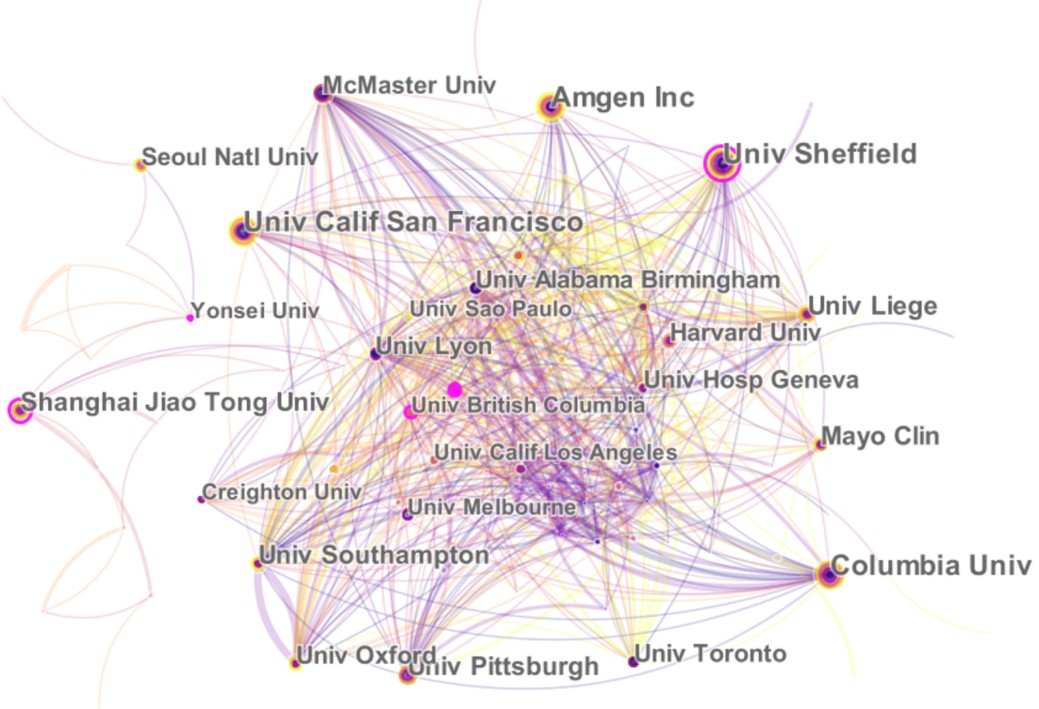

**Figure 4 The distribution of countries/regions and institutions.** The network map of institutions that involved in postmenopausal osteoporosis research.

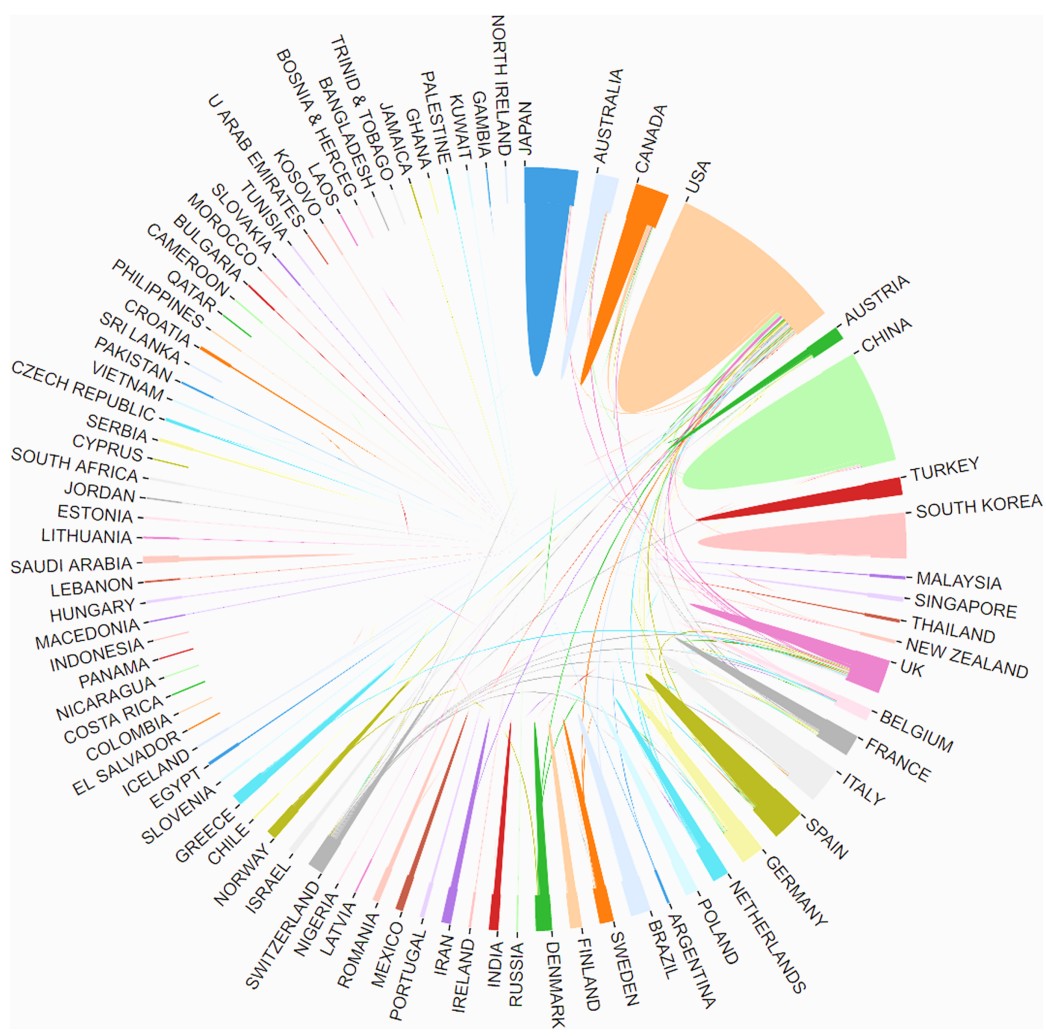

**Figure 5  The distribution of countries/regions and institutions.** The cooperation of countries/regions that involved in postmenopausal osteoporosis research.

*Journal of Bone and Mineral Research* has the largest IF of 5.711, followed by *Journal of Clinical Endocrinology and Metabolism* (5.605), *Bone* (4.36), *Osteoporosis International* (3.819), and *Maturitas* (3.654). According to the JCR 2018 standards, the top 10 most active journals were classified as Q1, sorted by the IF of the JCR category to which they belong.

## Distribution by author

Of all 19,615 authors included in this subject, the top 10 most productive authors engaged in related research were ranked by the number of published papers. They included Reginster JY, Cooper C, Kanis JA, Lewiecki EM, Rizzoli R, and Eastell R (Table 3). Among them, Reginster JY, from the Department of Public Health, Epidemiology and Health Economics, University of Liège in the Belgium, ranked first with 62 literatures, followed by Cooper C from Nuffield Department of Orthopaedics, Rheumatology and Musculoskeletal Sciences, University of Oxford in UK with 51 literatures. These two scholars made great

**Table 2 The top 10 most active journals that published articles in postmenopausal osteoporosis research (sorted by count).**

| Rank | Journal title | Article counts | Percentage (N/5,247) | IF(2018) | Quartile in category (2018) | H-index | Total number of citations | Average number of citations |
|------|---------------|----------------|----------------------|----------|------------------------------|---------|---------------------------|------------------------------|
| 1 | OSTEOPOROSIS INTERNATIONAL | 497 | 9.47% | 3.819 | Q1 | 157 | 2175 | 4.38 |
| 2 | BONE | 238 | 4.54% | 4.36 | Q1 | 183 | 964 | 4.05 |
| 3 | JOURNAL OF BONE AND MINERAL RESEARCH | 224 | 4.27% | 5.711 | Q1 | 223 | 1690 | 7.54 |
| 4 | CALCIFIED TISSUE INTERNATIONAL | 123 | 2.34% | 3.265 | Q1 | 106 | 332 | 2.7 |
| 5 | JOURNAL OF BONE AND MINERAL METABOLISM | 121 | 2.31% | 2.31 | Q1 | 66 | 218 | 1.8 |
| 6 | PLOS ONE | 120 | 2.29% | 2.776 | Q1 | 176 | 179 | 1.49 |
| 7 | JOURNAL OF CLINICAL ENDOCRINOLOGY & METABOLISM | 115 | 2.19% | 5.605 | Q1 | 98 | 778 | 6.77 |
| 8 | JOURNAL OF CLINICAL DENSITOMETRY | 86 | 1.64% | 2.184 | Q1 | 29 | 232 | 2.7 |
| 9 | MATURITAS | 81 | 1.54% | 3.654 | Q1 | 91 | 291 | 3.59 |
| 10 | MENOPAUSE-THE JOURNAL OF THE NORTH AMERICAN MENOPAUSE SOCIETY | 81 | 1.54% | 2.942 | Q1 | 93 | 189 | 2.33 |

achievements and are authorities in the research of postmenopausal osteoporosis. CiteSpace analyzed the information cited by the authors and co-cited authors, visualizing it in a network (Figs. 6 and 7). Kanis JA, with 1,374 co-citations, ranked first among the top 10 co-cited authors (Table 3), followed by Cummngs SR (991), Black DM (760), and Anonymous (687). These experts conducted a great quantity of research and laid a foundation for the development of the field of postmenopausal osteoporosis. The centrality of the first four authors was more than 0.1, indicating that they had formed an influential core scholar group in the domain of postmenopausal osteoporosis research.

## RESEARCH HOT SPOTS OF POSTMENOPAUSAL OSTEOPOROSIS

In the literature included, 2,439 major MeSH terms/MeSH subheading terms were computed with an accumulated frequency of 9,372 times. After H index standard evaluation, with an appearance of more than 36 times, a major MeSH term/MeSH subheading term was defined as being extremely frequent. Thirty-six extremely frequent major MeSH terms/MeSH subheading terms extracted from the included publications with an accumulated percentage of 38.22% (3,582/9,372) are displayed in Table 4. Different numbers of clusters were found by biclustering. Mountain visualization and matrix visualization showed the biclustering result of the matrix of source literatures - extremely frequent major MeSH terms/MeSH subheading terms. Mountain visualization and the extremely frequent major MeSH terms/MeSH subheading terms in each cluster

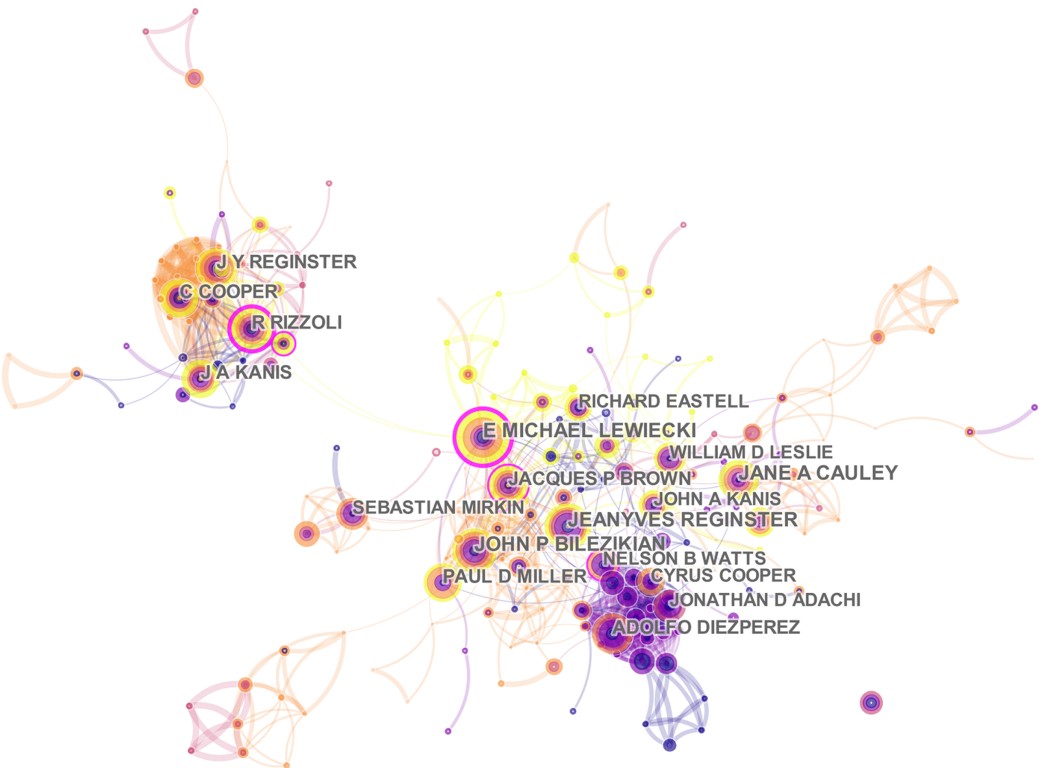

**Figure 6 The distribution of authors engaged in postmenopausal osteoporosis research.** The network map of productive authors.

classified into eight clusters are illustrated in Fig. 8. The intention of mountain visualization is to visually show the result of biclustering and the essence of high-dimensional datasets. Figure 8 displays each cluster as a peak in the 3D landform marked with the cluster number (from zero to 7, a total of eight clusters). The information about the associated cluster was reflected by its location on the plane, altitude, color and volume of its peak. When compared to other peaks, the location on the plane is the most informative attribute of a peak. The relative similarity of clusters is represented by the interval between peaks on the plane. The altitude of a peak is often in direct proportion to the internal similarity of the cluster. The internal standard deviation of objects in each cluster is revealed by the color of each peak. Blue means high deviation, while red means low deviation. Finally, the volume of a peak is in direct proportion to the amount of extremely frequent major MeSH terms/MeSH subheading terms stored within the cluster. Based on the authors' knowledge, a minimum of 30 publications should be contained in each independent cluster and triplet peaks should not appear in the mountain visualization. Figure 9 illustrates the matrix visualization, where the column tags are PMIDs of source literatures, and the row tags are extremely frequent major MeSH terms/MeSH subheading terms, separated on the bottom right of the matrix. The values present in the matrix are graphically represented by colors. The color of each reseau paints the proportional emergence frequency of a major MeSH term/MeSH subheading term in a publication. The cumulatively deeper red indicates greater significance, while the white
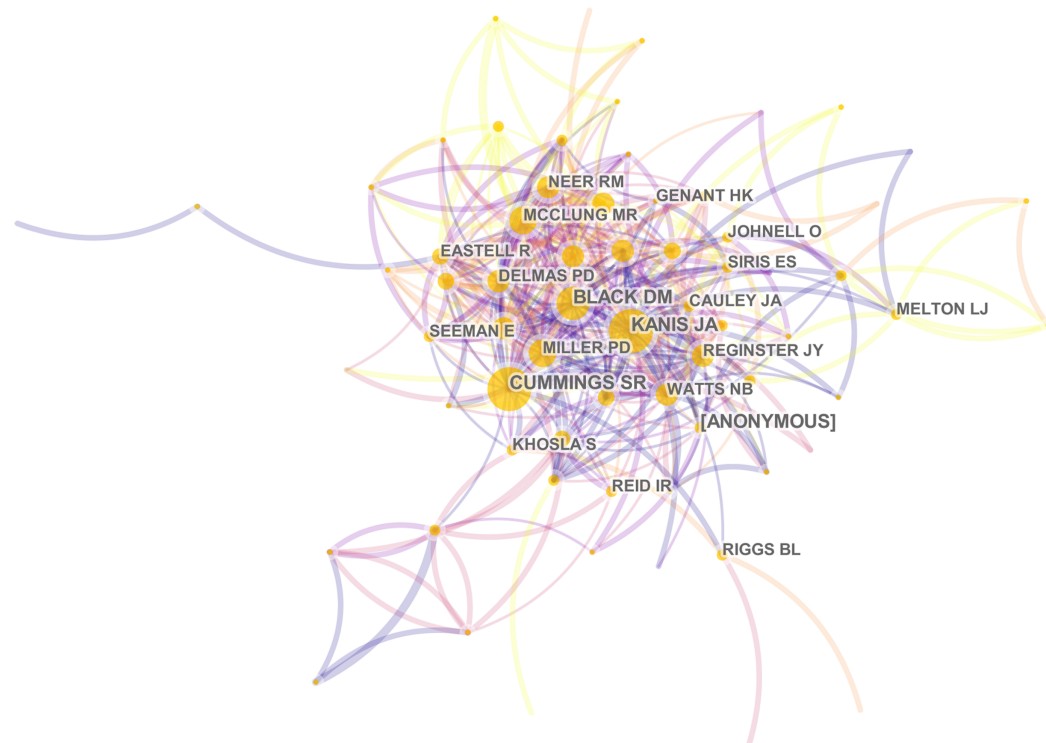

**Figure 7** **The distribution of authors engaged in postmenopausal osteoporosis research.** The network map of co-cited authors.                                   

**Table 3** **The top 10 most productive authors and co-cited authors contributed to publications in postmenopausal osteoporosis research.**

| Rank | Author | Article counts | Centrality | Total number of citations | Average number of citations | First author counts | First author citation counts | Average first author citation counts | Corresponding author | Corresponding author citation counts | Co-cited author | Citation counts | Centrality |
|------|--------|----------------|------------|---------------------------|----------------------------|---------------------|------------------------------|--------------------------------------|----------------------|--------------------------------------|-----------------|-----------------|------------|
| 1 | Reginster, JY | 62 | 0.01 | 810 | 13.06 | 10 | 57 | 5.7 | 13 | 75 | Kanis JA | 1374 | 0.37 |
| 2 | Cooper, C | 51 | 0.00 | 660 | 12.94 | 1 | 13 | 13 | 11 | 75 | Cummngs SR | 991 | 0.27 |
| 3 | Kanis, JA | 46 | 0.02 | 841 | 18.28 | 7 | 256 | 36.57 | 11 | 279 | Black DM | 760 | 0.16 |
| 4 | Lewiecki, EM | 44 | 0.38 | 505 | 11.48 | 10 | 45 | 4.5 | 19 | 74 | Anonymous | 687 | 0.02 |
| 5 | Rizzoli, R | 42 | 0.22 | 456 | 10.86 | 10 | 102 | 10.2 | 13 | 117 | Johnell O | 534 | 0.02 |
| 6 | Eastell, R | 39 | 0.07 | 280 | 7.18 | 5 | 29 | 5.8 | 7 | 37 | Mcclung MR | 507 | 0.08 |
| 7 | Adachi, JD | 38 | 0.03 | 181 | 4.76 | 1 | 1 | 1 | 1 | 1 | Khosla S | 483 | 0.01 |
| 8 | Lee, SH | 38 | 0.00 | 90 | 2.37 | 5 | 8 | 1.6 | 7 | 33 | Reginster, JY | 449 | 0.04 |
| 9 | Brandi, ML | 37 | 0.11 | 368 | 9.95 | 2 | 3 | 1.5 | 11 | 16 | Rjggs BL | 438 | 0.10 |
| 10 | Miller, PD | 36 | 0.00 | 306 | 8.5 | 9 | 80 | 8.89 | 10 | 96 | Reid IR | 428 | 0.03 |

indicates the significance is closer to none. In Table 5, gCLUTO replumed the rows of the initial matrix so that analogous rows in the same cluster are converged; these clusters are partitioned by black horizontal lines. Thirty-six extremely frequent major MeSH terms/MeSH subheadings terms were clustered into eight clusters in the matrix visualization. The top layered cluster tree describes the relationships among literatures,

**Table 4 Highly frequent major MeSH terms from the included publications on postmenopausal osteoporosis (n = 9372).**

| Rank | Major MeSH terms/ MeSH subheadings | Frequency | Proportion of frequency (%) | Cumulative percentage (%) |
|---|---|---|---|---|
| 1 | Osteoporosis, Postmenopausal/drug therapy | 577 | 6.1566 | 6.1566 |
| 2 | Bone Density Conservation Agents/therapeutic use | 305 | 3.2544 | 9.411 |
| 3 | Osteoporosis, Postmenopausal/prevention & control | 208 | 2.2194 | 11.6304 |
| 4 | Bone Density/drug effects | 185 | 1.974 | 13.6044 |
| 5 | Bone Density | 172 | 1.8353 | 15.4396 |
| 6 | Bone Density Conservation Agents/administration & dosage | 135 | 1.4405 | 16.8801 |
| 7 | Bone Density/physiology | 116 | 1.2377 | 18.1178 |
| 8 | Osteoporotic Fractures/prevention & control | 113 | 1.2057 | 19.3235 |
| 9 | Osteoporosis, Postmenopausal/epidemiology | 110 | 1.1737 | 20.4972 |
| 10 | Osteoporosis, Postmenopausal/complications | 104 | 1.1097 | 21.6069 |
| 11 | Diphosphonates/therapeutic use | 102 | 1.0883 | 22.6953 |
| 12 | Osteoporosis, Postmenopausal/genetics | 96 | 1.0243 | 23.7196 |
| 13 | Osteoporosis, Postmenopausal/diagnosis | 94 | 1.003 | 24.7226 |
| 14 | Osteoporosis, Postmenopausal/metabolism | 94 | 1.003 | 25.7256 |
| 15 | Postmenopause | 92 | 0.9816 | 26.7072 |
| 16 | Osteoporosis, Postmenopausal/blood | 91 | 0.971 | 27.6782 |
| 17 | Osteoporosis, Postmenopausal/diagnostic imaging | 80 | 0.8536 | 28.5318 |
| 18 | Osteoporosis, Postmenopausal/physiopathology | 72 | 0.7682 | 29.3 |
| 19 | Osteoporosis, Postmenopausal/therapy | 68 | 0.7256 | 30.0256 |
| 20 | Bone Density Conservation Agents/adverse effects | 67 | 0.7149 | 30.7405 |
| 21 | Osteoporotic Fractures/epidemiology | 53 | 0.5655 | 31.306 |
| 22 | Bone Remodeling/drug effects | 51 | 0.5442 | 31.8502 |
| 23 | Bone Density Conservation Agents/pharmacology | 50 | 0.5335 | 32.3837 |
| 24 | Osteoporosis/drug therapy | 50 | 0.5335 | 32.9172 |
| 25 | Bone and Bones/drug effects | 48 | 0.5122 | 33.4294 |
| 26 | Osteoporosis, Postmenopausal/pathology | 47 | 0.5015 | 33.9309 |
| 27 | Diphosphonates/administration & dosage | 46 | 0.4908 | 34.4217 |
| 28 | Osteoporosis, Postmenopausal/etiology | 43 | 0.4588 | 34.8805 |
| 29 | Bone Density/genetics | 43 | 0.4588 | 35.3393 |
| 30 | Osteoporotic Fractures/etiology | 41 | 0.4375 | 35.7768 |
| 31 | Dietary Supplements | 40 | 0.4268 | 36.2036 |
| 32 | Alendronate/therapeutic use | 40 | 0.4268 | 36.6304 |
| 33 | Diphosphonates/adverse effects | 39 | 0.4161 | 37.0465 |
| 34 | Postmenopause/physiology | 37 | 0.3948 | 37.4413 |
| 35 | Bone and Bones/metabolism | 37 | 0.3948 | 37.8361 |
| 36 | Teriparatide/therapeutic use | 36 | 0.3841 | 38.2202 |

and the left layered cluster tree demonstrates the relationships among extremely frequent major MeSH terms/MeSH subheading terms. Each cluster also shows which of the major MeSH term/MeSH subheading terms exists in matching literatures. A deeper exploration of the typical literatures in each cluster was conducted to discern between and to outline the themes of each cluster. According to the standards discussed above by the

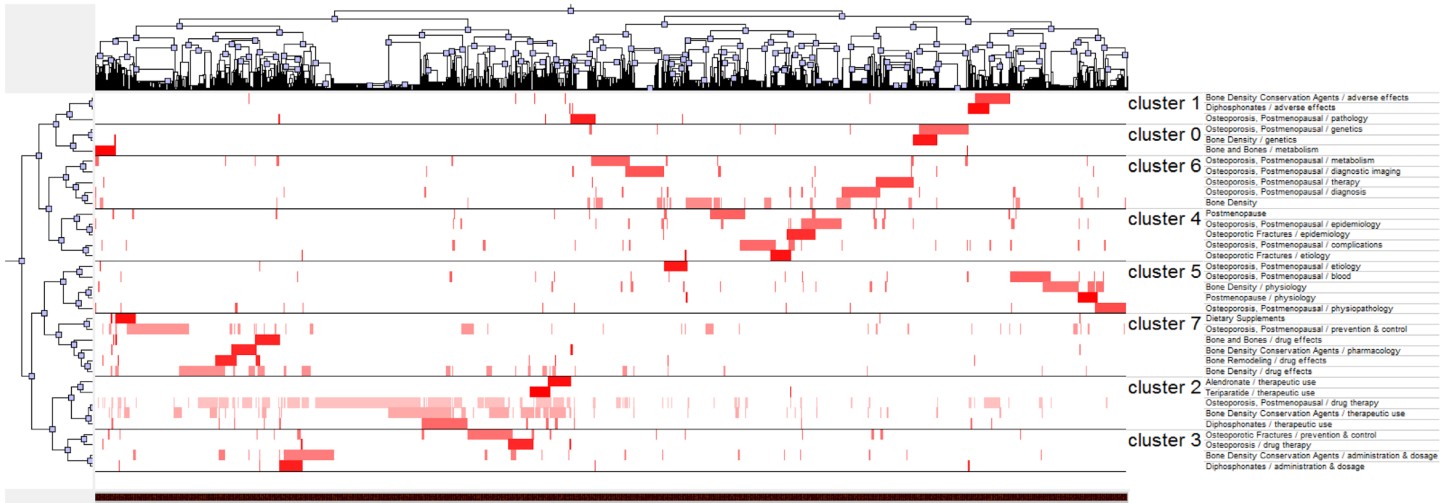

**Figure 8 Mountain visualization of biclustering of highly frequent major MeSH terms and articles on postmenopausal osteoporosis.**

**Figure 9 Visualized matrix of biclustering of highly frequent major MeSH terms and PubMed Unique Identifiers (PMIDs) of articles on postmenopausal osteoporosis.**

**Table 5 Highly frequent major MeSH a terms-source articles matrix (localized).**

| No. | Major MeSH terms/ MeSH subheadings | Pubmed Unique Identifiers of source articles | | | | |
|---|---|---|---|---|---|---|
| | | 21631599 | 22057139 | 22302614 | … | 29782125 |
| 1 | Osteoporosis, Postmenopausal/ drug therapy | 0 | 0 | 0 | … | 0 |
| 2 | Bone Density Conservation Agents/therapeutic use | 0 | 0 | 0 | … | 0 |
| 3 | Osteoporosis, Postmenopausal/ prevention & control | 1 | 0 | 1 | … | 0 |
| 4 | Bone Density/drug effects | 0 | 0 | 1 | … | 0 |
| … | … | … | … | … | … | … |
| 35 | Bone and Bones/metabolism | 0 | 0 | 0 | … | 0 |
| 36 | Teriparatide/therapeutic use | 0 | 0 | 0 | … | 0 |

research team, the major MeSH terms/MeSH subheading terms were categorized into eight clusters (Fig. 8). These clusters include:

Genetics-related research on bone metabolisms of postmenopausal osteoporosis (Cluster 0),

Adverse effects of diphosphonates (Cluster 1),

Therapeutic treatment of postmenopausal osteoporosis (Cluster 2),

Administration and dosage of Clinical therapy drug——diphosphonates (Cluster 3),

Study on epidemiology and etiology of complications of postmenopausal osteoporosis (Cluster 4),

Physiology and physiopathology of postmenopausal osteoporosis (Cluster 5),

Risk factors associated with bone mineral density (BMD) in the diagnosis of postmenopausal osteoporosis (Cluster 6),

Clinical drug effects of dietary supplements on postmenopausal osteoporosis (Cluster 7).

## DISCUSSION

According to the statistical and quantitative analysis by the Online Analysis Platform of Literature Metrology, software CiteSpace and BICOMB, the research output on postmenopausal osteoporosis has gradually decreased over the past five years. MeSH terms can represent the content of literatures and a great quantity of MeSH terms can map the current research status and trends of the field. According to a qualitative and co-word biclustering analysis by gCLUTO software, similar MeSH terms can be identified and categorized into clusters. This is how the research hot spots on postmenopausal osteoporosis were generated, making the essential knowledge structure and trends in this field able be examined systematically.

Cluster 0 relates to genetic research on bone metabolisms of postmenopausal osteoporosis. Postmenopausal osteoporosis is a common polygenic bone metabolic disease. Genetic factors play an important role in the bone metabolism regulation of postmenopausal osteoporosis. BMD, a crucial risk factor for osteoporosis, is highly genetic

with estimates of heritability ranging from 0.5 to 0.9. To date, several studies have reported that some functional genes, such as CYP11A1 in vitamin D and estrogen hormone-response pathways, the estrogen receptorα (ERα) gene, tumor necrosis factor (TNF)-α gene, and TNFSF11, TNFRSF11A in the RANKL/RANK/OPG pathway, are implied to be associated with BMD in postmenopausal osteoporosis (*Tu et al., 2015*). Exploring different genetic variants underlying the development of osteoporosis would make the early prophylactics of osteoporosis possible, as well as the ability to manage individual-based symptomatic treatment.

Cluster 1 relates to adverse effects of diphosphonates. For the treatment of osteoporosis, the most widely used medications are diphosphonates, which are divided into two groups on the basis of their structures. First generation diphosphonates do not contain nitrogen, while new generation diphosphonates have a nitrogen-containing side chain. This structure has a high-affinity for hydroxyapatite at the bone surface, so diphosphonates can preserve for months or even years. After years of evolution, diphosphonates, which include alendronate (ALN), risedronate sodium, ibandronate sodium (IBN) and zoledronic (ZOL), are more durable and stable. The adverse effects of diphosphonates, however, are unavoidable. The first intravenous dose of diphosphonates like IBN and ZOL may trigger an acute-phase response (APR) where after their first diphosphonate infusion, patients have had fevers and pains. Commonly, these symptoms were transient in duration and mild to moderate in intensity, and according to NSAID, the incidence and intensity of such an APR could be efficiently impeded (*Ding et al., 2017*). Additionally, the atypical femoral fracture (AFF), an unusual atraumatic or minimal-trauma fracture, has also been reported with increasing frequency in long-term diphosphonate users since the first case reports were published in 2005 (*Kim et al., 2015*). A unique case of AFF after diphosphonate therapy was discovered in 2014, but the patient had a successful recovery through conservative treatment (*Pazianas & Smith, 2014*). To summarize, it is essential to assess the possibility of atypical fractures in osteoporotic patients when they complain about lower extremity pain, and to take into account alternative treatments instead of diphosphonates.

Cluster 2 relates to the therapeutic treatment of postmenopausal osteoporosis. Drug therapy for osteoporosis can be divided into antiresorptive agents and anabolic agents. Antiresorptive agents are composed of raloxifene (RAL), diphosphonates, and denosumab. Teriparatide is the only anabolic agent for osteoporosis treatment approved by the Food and Drug Administration. Studies have shown that cortical turnover and cortical bone formation in patients who were either treatment naïve (TN) or had previous ALN therapy increased with 24 months of teriparatide treatment (*Ma et al., 2014*). In other clinical studies, synergistic effects of combination therapy with an antiresorptive agent and teriparatide have been proposed (*Shen, Gray & Martinez, 2017*). The addition of ALN to ongoing teriparatide treatment, and continuing ALN after teriparatide was stopped may be beneficial for patients in terms of areal and volumetric BMD increase (*Muschitz et al., 2014*). Furthermore, the treatment of combining teriparatide with diphosphonates has shown faster bony unions and highly improved BMD scores (*Cho et al., 2017*).

Although combination therapy has obvious advantages, the best time to start combination therapy should be further studied in order to prevent osteoporotic fractures.

Cluster 3 relates to the administration and dosage of diphosphonates. Diphosphonates as an anti-resorptive agent have been accepted for the treatment and prevention of postmenopausal osteoporosis. However, official guidance on the dosage and the length of treatment is lacking, and the curative effect of diphosphonates is not ideal. First, long-term users with 10 dose years or more of a diphosphonates are rare due to periods of low compliance and gaps, with a discrepancy between the length of treatment and doses taken (*Abrahamsen, 2013*). Second, long-term diphosphonate treatment in postmenopausal women does not impair the response to subsequently administered intravenous pamidronate, suggesting that the inadequate response to long-term diphosphonate treatment is not responsible for treatment failure (*Yavropoulou, Hamdy & Papapoulos, 2013*). What's more, over the past decade, several reports have highlighted the increased risk of AFF in patients treated with long-term diphosphonates. On the basis of this recommendation, patients may be advised to stop taking diphosphonates for a while. Total hip BMD declines significantly within 1 year of discontinuing diphosphonates, particularly in lean patients (*Xu et al., 2016*). Cluster 1 has narrated the side effects of diphosphonates, and additional studies are needed to identify reasonable treatments using diphosphonates.

Cluster 4 relates to the epidemiology and etiology of complications of postmenopausal osteoporosis. The worst complications of postmenopausal osteoporosis are fractures, so the accurate assessment and prediction for the risk of fractures are particularly crucial. DXA had been regarded worldwide as the gold standard for the diagnosis of osteoporosis at the lumbar spine and hip, but BMD reveals only a portion of an individual's fracture risk because of the multi-factor fragility fracture. Additionally, to identify patients with a high risk of fracture, many clinical risk factors must be taken into consideration as well as BMD, increasing the possibility of osteoporotic fractures for high-risky patients. The Fracture Risk Assessment Tool (FRAX), uses nine clinical risk factors to predict an individual 10-year risk of major or hip osteoporotic fractures: age, sex, BMI, prior fragility fracture history, family history of hip fracture, the existence of secondary osteoporosis, exposure to systemic glucocorticoids, current smoking and three or more units of alcohol per day. In addition, the International Osteoporosis Foundation (IOF) One Minute Test, though with the lowest predicting rate when compared to other tested tools, has shown competent prediction precision (*Briot et al., 2013*; *Kharroubi et al., 2017*). Moreover, there is an increased risk for hip fracture in postmenopausal women with type 2 diabetes (*Dytfeld & Michalak, 2017*). Further etiology studies should be conducted to prevent the occurrence of the complications discussed above.

Cluster 5 relates to the physiology and physiopathology of postmenopausal osteoporosis. A strong correlation between BMD scores and the probability of fragility fractures has been well-documented. BMD is affected by multiple factors. Higher BMI scores and moderate levels of physical activity have been found significant in avoiding a decline of BMD (*Wee et al., 2013*). Life satisfaction and BMD improvement are longitudinally linked with reduced bone loss in postmenopausal women (*Rauma et al., 2014*).

Cluster 6 relates to risk factors associated with BMD in the diagnosis of postmenopausal osteoporosis. With the increasing incidence of postmenopausal osteoporosis, it is important to identify risk factors associated with BMD for the prevention of postmenopausal osteoporosis. As there are many factors causing postmenopausal osteoporosis, it is difficult to accurately pinpoint its risk factors. Exercise is consistently effective in (initially) favorably affecting BMD in early-postmenopausal women without any leveling-off effect after 16 years of exercise (*Kemmler, Engelke & Von Stengel, 2016*). Duration of fertility (years of menstruation) longer than 33 years and a BMI greater than 32 seem to prevent postmenopausal osteoporosis. Age is also an independent risk factor for postmenopausal osteoporosis (*Cavkaytar et al., 2015*). When it comes to diagnostic imaging, probabilistic sensitivity analysis, DXA and quantitative CT at 55 years-old with quantitative CT screening every 5 years was the best strategy. Furthermore, a combined assessment of bone strength and BMD is a cost-effective strategy for osteoporosis screening in postmenopausal women and has the potential to prevent a large number of osteoporosis fractures.

Cluster 7 relates to the drug effect of alternative therapy—dietary supplements. Pharmacotherapy, diphosphonates for instance, has been widely used to alleviate the risk of fractures and remedy osteoporosis. With low compliance and related adverse effects associated with long-term medication, it is crucial to develop new alternative medicine to treat osteoporosis. Additionally, many people desire alternative and supplemental therapies. A calcium collagen chelate (CC) dietary supplement has shown to be effective in improving BMD and blood biomarkers of bone turnover in osteopenic postmenopausal women (*Castelo-Branco, 2015*; *Elam et al., 2015*). Context Eucommiae Cortex and Radix Dipsaci, occurring in a ratio of 1:1 in Du-Zhong-Wan (DZW) and Puerarin 600-O-xyloside, also achieved the same effect as above on ovariectomy mice (*Li et al., 2016a*, *2016b*). These have provided new ways to treat patients with osteoporosis.

Nonetheless, we realize several latent limitations in this study. First, although co-word biclustering, based on extremely frequent MeSH terms, is a highly beneficial way to determine research hot spots in a field, the number of MeSH terms might have some effect on the biclustering analysis results (although the updated emerging themes with low attention may not have been involved). Second, the database updates research continuously, so there may be a discrepancy between bibliometric analysis data and real study conditions, and the number of PMOP papers may grow rapidly with future research breakthroughs.

## CONCLUSIONS

Our study found etiology and medication as key points in postmenopausal osteoporosis research. Epidemiology studies developed BMD and FRAX to predict the individual risk of osteoporotic fracture, to summarize high-risk factors associated with PMOP, and to discern between key genes or microenvironmental factors related to PMOP. All these studies laid the foundation of basic research, especially in terms of genetics. Another hot field is drug treatment. After many years of randomized controlled trials (RCT), current anti-osteoclastogenesis drugs and their side effects have been surveyed and evaluated in detail. Teriparatide and some novel medicines with higher efficacy in

promoting osteogenesis should be paid more attention from experts and scholars, and dietary supplements would actually be excellent substitutes for drugs because of their accessibility and low toxicity. The aforementioned hot spots might see great scientific breakthroughs in the near future, and our research might reflect a new direction for postmenopausal osteoporosis research.

### Funding
This study was supported by the National Natural Science Foundation of China (Grant No. 81472044, 81271939). The funders had no role in study design, data collection and analysis, decision to publish, or preparation of the manuscript.

### Grant Disclosures
The following grant information was disclosed by the authors:
National Natural Science Foundation of China: 81472044, 81271939.

### Competing Interests
The authors declare that they have no competing interests.

### Author Contributions
- Siming Zhou conceived and designed the experiments, performed the experiments, analyzed the data, prepared figures and/or tables, authored or reviewed drafts of the paper, approved the final draft.
- Zhengbo Tao analyzed the data, prepared figures and/or tables, authored or reviewed drafts of the paper, approved the final draft.
- Yue Zhu conceived and designed the experiments, contributed reagents/materials/analysis tools, authored or reviewed drafts of the paper, approved the final draft.
- Lin Tao conceived and designed the experiments, contributed reagents/materials/analysis tools, authored or reviewed drafts of the paper, approved the final draft.

### Data Availability
The raw data is available in the Supplemental Files.

### Supplemental Information
Supplemental information for this article can be found online at http://dx.doi.org/10.7717/peerj.8145#supplemental-information.

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
