# Peer review of "Mapping theme trends and recognizing hot spots in postmenopausal osteoporosis research: a bibliometric analysis"

_PeerJ, doi:10.7717/peerj.8145_

## Round 0.1 · original submission · Major Revisions

Your manuscript has been reviewed and requires modifications prior to making a decision. The comments of the reviewer(s) are included at the bottom of this letter.

Reviewers indicated that the novelty of the study, discussion, conclusion, and references should be reviewed and the manuscript should be improved. The manuscript also needs extensive English editing because there are several typos and grammatical errors. I agree with this evaluation and I would, therefore, request for the manuscript to be revised accordingly. I would also like to suggest the following changes:

Provide more details according to the co-word biclustering analysis.

In addition to this, the similarity index in the Turnitin Originality Report (I also attached) was 31%, and this value exceeded the acceptable level. Authors must be careful about citing papers appropriately in their study. I am not an expert in this field, but these studies below are the most three similar papers according to the results of the Turnitin Originality Report.

https://doi.org/10.2196/jmir.3326
https://doi.org/10.1038/nrdp.2016.70
https://doi.org/10.1186/s12886-018-0752-z

Reviewer 1 ·

Basic reporting

no comment

Experimental design

no comment

Validity of the findings

no comment

Additional comments

This bibliometric analysis presents the research progress on postmenopausal osteoporosis in
recent five years. It is a piece of "routine" work. The analysis needs to combine subject specific expert knowledge with careful bibliometric analysis, applying suitable bibliometric methods and tools. And the article should be clearly written, and the results also should be presented and discussed adequately to provide interesting information for the relevant scientific community from an innovative perspective.

I have several concerns about the paper:

1. Little innovation was shown. The authors should state what is new in this study. The novelty of the study should be added to the manuscript.
2. Introduction should be improved. For example, in line 74-76, “While the above reviews to a certain extent, reflects the research status on postmenopausal osteoporosis, different scholars have different research contents and views in this field, and lack of long-term bibliometric research.” Do you think five years’ bibliometric analysis represents a long-term research? How you think bibliometric analysis on postmenopausal osteoporosis in recent five years can presents the research progress and theme trends? Why do you only choose the recent five years as the study period?
3. Line 82-105 should be appeared in the literature review section. This paragraph only summarized paper published using biclustering method. What can these statements make sense in your paper?
4. Line 116. “Data collection”. “PubMed, providing free access to MEDLINE, OLDMEDLINE, and other related databases and developed by the national center for biotechnology information (NCBI) of the national library of medicine (NLM), from which related data were extracted and downloaded without the restriction of language.” There are some other comprehensive ones, such as "Scopus", “WOS” available. Authors should explain why PubMed was the chosen one rather than others or include a broader set of databases in the study.
5. Results and Discussion. Authors simply listed some statistical information about Growth of the relevant Literature, Distribution characteristics of literature, Most Active Journals, which will not help. Moreover, most of the discussion seems like an explanation of the figures.
6. The conclusion is too simple, and it is just a repeat of the research purpose.
7. The presentation of the figures is not good with Low image quality.

Annotated reviews are not available for download in order to protect the identity of reviewers who chose to remain anonymous.

Reviewer 2 ·

Basic reporting

This article is in need of editing for English usage. Many sentences require rewriting to clarify their exact meaning. For example, "The health of the elderly has received more and more attention this year." (Line 39) [Do the authors mean "in recent years"?]; "MeSH terms can image the content of articles and a great quantity" (Line 253); "To summarize, it is essential to assess the possibility of atypical fractures in osteoporotic patients when they beef about" (Line 290); In addition, the use of capital letter and small letter requires attention. For example, "and developed by the national center for biotechnology information (NCBI) of the national library of medicine (NLM)," (Line 119); "Additional Studies on postmenopausal" (Line 388).

The reference style needs to conform to the usual convention. For example, only author's last name but not initial is needed when referring to a publication. For example, Line 92, 101, and 102.

A few citations seem to be out of place. For example, reference [3] should be placed at the end of Line 56 instead of 51.

The reference for the sentence "Yu, Z et al characterized the important research topics in cancer discovery using gene expression profiles" (Line 92) is missing.

The authors mentioned "To the best knowledge of the authors, for the time being, there have been few bibliometric articles on postmenopausal osteoporosis.". Please provide references for and a brief descriptive of these articles.

The authors mentioned "A FRAX,... has been developed and acknowledged by the WHO to predict an individual 10-year risk of major or hip osteoporotic fractures.". According to a Bulletin of the World Health Organization (https://www.who.int/bulletin/volumes/94/12/16-188532/en/): "FRAX® tool to evaluate fracture risks of patients is not a “WHO tool” and has not been developed, endorsed, evaluated or validated by WHO". Please revise the sentence accordingly.

Experimental design

Methods has provided sufficient detail & information to replicate.

Validity of the findings

Conclusions are well stated, linked to original research question & limited to supporting results.

The Discussion section will benefit if the authors can, based on their findings, identify and speculate what are the needed but missing area of postmenopausal osteoporosis research.

Additional comments

No comment.

Reviewer 3 ·

Basic reporting

The authors should minimize long sentences

Experimental design

No comment

Validity of the findings

No comment

Additional comments

Comment
The text is well written and the review of postmenopausal osteoporosis literature seems to represent quite adequately the temporal development and evolution of these studies. However, few minor specific corrections has been made in the annotated pdf while the general comments are below:
Result
1. In table 1, the authors may add the impact factor (If available) of the journals and country of affiliation. This will help correlate if they also align with the top productive countries.
Discussion
1. The authors may interpret the observation of publication of ≥20 articles per author. It is an indication of specialized area of research. Hence, the absence of transient authors who publish sporadically.
2. The authors should briefly discuss the top journal. These journals are potentially publication niche for future breakthrough in postmenopausal osteoporosis research and should be watched.
3. English being official language of England and USA may be a contributing factor of the high publications, but I seriously doubt if that is the major reason. Major reason: USA and UK (England) have a high gross domestic expenditure on research and development (R&D) as published by World Bank. USA has gross expenditure on R&D above the Organization for Economic Co-operation and Development (OECD) average of 2.3% with UK close by. A country’s scientific output is a reflection of its earlier investment in research and development. This is evident in many bibliometric studies as cited in the article.
4. The authors should minimize long sentences

Annotated reviews are not available for download in order to protect the identity of reviewers who chose to remain anonymous.

---

## Round 0.2 · Minor Revisions

The manuscript has been assessed by three reviewers. Two of the three reviewers agree on the fact that there are still a few points that need to be addressed. Reviewers indicated that references in the manuscript should be corrected. We would be glad to consider a substantial revision of your work, where the reviewers' comments will be carefully addressed one by one.

Reviewer 1 ·

Basic reporting

no comment

Experimental design

no comment

Validity of the findings

no comment

Additional comments

Thank you for the considerable work in revising your paper. I found that my previous comments were addressed and noticed the hard work made for making the contents of the paper clear. I appreciated the effort and am therefore satisfied with the revised version.

Reviewer 2 ·

Basic reporting

The authors have adequately addressed all my comments except there are still more than a few errors in the Reference section.

Experimental design

Methods described has provided with sufficient detail & information to replicate.

Validity of the findings

The conclusions were appropriately stated and were connected to the original question investigated. They were limited to those supported by the results.

Additional comments

The authors have adequately addressed all my comments except there are still errors in the Reference section.
For example,
1. The volume and page number information is missing in "Chen C, Ibekwe‐Sanjuan F, Hou JJJotASfIS, and Technology. 2010. The structure and dynamics of cocitation clusters: A multiple‐perspective cocitation analysis."
2. There are errors in the authors' names in "Chen CJJotAfIS, and Technology. 2014. CiteSpace II: Detecting and visualizing emerging trends and transient patterns in scientific literature. 57:359-377."
3. Abbreviated title should be used in "Cui L LW YL, Zhang H, Hou YF, Huang YN. 2008. Development of a text mining system based on the co-occurrence of bibliographic items in literature. New Technology of Library and Information Service 8:70-75."
4. Sentence case instead of title case should be used in "Kemmler W, Engelke K, and von Stengel S. 2016. Long-Term Exercise and Bone Mineral Density Changes in Postmenopausal Women--Are There Periods of Reduced Effectiveness? J Bone Miner Res 31:215-222. 10.1002/jbmr.2608"

Reviewer 3 ·

Basic reporting

Review comment
The authors has greatly improved the manuscript. Just a minor correction:
1. In line 163, the authors wrote: In total, 5,247 literatures, 4,466 articles and 781 reviews (Fig.1).
I guess it should read: In total, 5,247 literatures comprising 4,466 articles and 781 reviews for clarity or the later part put in bracket as in figure 1.

2. In figure 1, the authors wrote: 5,247 articles identified (including 4,466 articles and 781 reviews.
3. Article was used twice. I believe the use of “literature” in the first instance as used in the result section would be better than “article”. Hence: 5,247 literatures identified (including 4,466 articles and 781 reviews).

Experimental design

Concerns have been addressed

Validity of the findings

Concerns have been addressed

Additional comments

Review comment
The authors has greatly improved the manuscript. Just a minor correction:
1. In line 163, the authors wrote: In total, 5,247 literatures, 4,466 articles and 781 reviews (Fig.1).
I guess it should read: In total, 5,247 literatures comprising 4,466 articles and 781 reviews for clarity or the later part put in bracket as in figure 1.

2. In figure 1, the authors wrote: 5,247 articles identified (including 4,466 articles and 781 reviews.
3. Article was used twice. I believe the use of “literature” in the first instance as used in the result section would be better than “article”. Hence: 5,247 literatures identified (including 4,466 articles and 781 reviews).

---

## Round 0.3 · accepted · Accept

The authors addressed the reviewers' concerns and substantially improved the content of MS.

So, based on my own assessment as an editor, no further revisions are required and the MS can be accepted in its current form.

Reviewer 1 ·

Basic reporting

no comment

Experimental design

no comment

Validity of the findings

no comment

Additional comments

The revisions are fine.

Reviewer 2 ·

Basic reporting

The authors have adequately addressed all my comments except one of the citation can be further improved.

Experimental design

Methods described has provided with sufficient detail & information to replicate.

Validity of the findings

The conclusions were appropriately stated and were connected to the original question investigated. They were limited to those supported by the results.

Additional comments

The authors have adequately addressed all my comments. One final suggestion, please fix the citation for the gCLUTO software. Currently, it is shown as "K 2014". The authors may want to consider showing the URL of the software (http://glaros.dtc.umn.edu/gkhome/cluto/gcluto/overview) along with its original publication "Rasmussen M & Karypis G. 2004. gcluto: An interactive clustering, visualization, and analysis system. UMN-CS TR-04, 21(7)."

Reviewer 3 ·

Basic reporting

Concerns have been resolved

Experimental design

Concerns have been resolved

Validity of the findings

N/A

Additional comments

Concerns have been resolved